# Schizophrenia more employable than depression? Language-based artificial intelligence model ratings for employability of psychiatric diagnoses and somatic and healthy controls

**Maximin Lange**[1]*, **Alexandros Koliousis**[2], **Feras Fayez**[3,4], **Eoin Gogarty**[1,3], **Ricardo Twumasi**[1]

1 Institute of Psychiatry, Psychology & Neuroscience, King's College London, London, United Kingdom, 2 Northeastern University, London, United Kingdom, 3 King's College Hospital NHS Foundation Trust, London, United Kingdom, 4 Imperial College Healthcare NHS Trust, London, United Kingdom

* Maximin.lange@kcl.ac.uk

## Abstract

Artificial Intelligence (AI) assists recruiting and job searching. Such systems can be biased against certain characteristics. This results in potential misrepresentations and consequent inequalities related to people with mental health disorders. Hence occupational and mental health bias in existing Natural Language Processing (NLP) models used in recruiting and job hunting must be assessed. We examined occupational bias against mental health disorders in NLP models through relationships between occupations, employability, and psychiatric diagnoses. We investigated Word2Vec and GloVe embedding algorithms through analogy questions and graphical representation of cosine similarities. Word2Vec embeddings exhibit minor bias against mental health disorders when asked analogies regarding employability attributes and no evidence of bias when asked analogies regarding high earning jobs. GloVe embeddings view common mental health disorders such as depression less healthy and less employable than severe mental health disorders and most physical health conditions. Overall, physical, and psychiatric disorders are seen as similarly healthy and employable. Both algorithms appear to be safe for use in downstream task without major repercussions. Further research is needed to confirm this. This project was funded by the London Interdisciplinary Social Science Doctoral Training Programme (LISS-DTP). The funders had no role in study design, data collection and analysis, decision to publish, or preparation of the manuscript.

## 1. Introduction

### 1.1 Background

Until recently, human resource (HR) professionals would manually sift through vast amounts of resumes to determine the best fitting candidate for a job opening. Judgement calls were made based on their own experience, gut feeling, or discussions with colleagues.

**Data Availability Statement:** All relevant data are within the manuscript and its Supporting information files.

**Funding:** This project was funded by the London Interdisciplinary Social Science Doctoral Training Programme (LISS-DTP).

**Competing interests:** N/A.

Artificial intelligence (AI) has revolutionised this process, assisting in extracting information about skills of applicants from resumes, social media sites and cover letters [1–3] a practice known as resume parsing. AI models that can parse resumes are so widespread that virtually all Fortune 500 firms today use some version of them [4].

Similarly, individuals in the market for a new job had to look up openings manually. Especially online, job seekers would face a plethora of available vacancies. A scenario of potentially overwhelming nature.

Job seekers now receive help by AI-powered job recommender systems (JRS). Job recommendation system are powered by machine learning algorithms, bringing together vacancies and job seekers based on behaviours, preferences or needs of the two parties [5].

Despite the excitement surrounding AI and JRS in HR, as it eases lives of recruiters and job seekers alike, such systems ought to be assessed in a broader spectrum of existing organisational transformation.

## 1.2 Occupational bias and allocation harm

There is a rich history of bias in general recruiting against a broad spectrum of attributes, including all protected characteristics, which have been extensively studied and summarised elsewhere [6–8].

Although more recent research suggests a changing trend [9], there is ample evidence for discrimination against people with mental health disorders in recruiting [10–13].

This leaves healthy people and people with mental illness, who might possess equivalent levels of skills, being employed at different rates. As a result, we might see job databases labelling mentally healthy people as more employable, and safe to work with.

Contrary to stigma, most people with mental health disorders report a desire to work [14–16].

It is often forgotten that people with mental illness still possess knowledge, skills, and abilities that can facilitate organisational effectiveness [17] and the World Health Organization [18] stresses that under-use of skills in the workplace can aggravate mental health disorders.

## 1.3 Natural language processing

Natural Language Processing (NLP) encompasses a collection of methods that transform written text passages into datasets that can subsequently be analysed using traditional statistics and machine learning models [19, 20]. As an interdisciplinary field, NLP bridges artificial intelligence, computer science, cognitive science, information processing, and linguistics, letting computers parse and process human language [21].

Real-time applications of NLP in the business field include chatbots, sentiment analysis, and speech recognition [22–25]. In the healthcare and pharmaceutical industries, NLP is used to analyse large amounts of unstructured data, such as electronic health records, to gather insights into patient behaviour, disease synthesis and prognostic predictions [26–30]. Additionally, NLP can be integrated with qualitative research methods to augment traditional text analysis approaches [20, 31, 32] include machine translation [33], or information retrieval and question answering [34, 35].

In the following two sections we address challenges related to occupational bias in NLP when used in the context of recruiting and job searching.

**1.3.1 Recruiting and occupational bias in NLP.** For recruiting purposes, NLP can be trained, among other things, on resumes, cover letters and social media profiles or posts [36–39]. At this stage, such data is already influenced by an individual's personal background.

Language use is heavily affected by, and dependent on, culture, age, gender and personality [40–43]. Specific word use patterns of people with mental illness have been identified: Perceptual and causal language are negatively correlated in schizophrenic individuals but positively correlated in those with mood disorders [44]. Spoken language patterns might even allow for prediction of psychosis [45, 46]. Detection of general mental illnesses and even reliable estimation of population mental health is possible through analyses of social media posts, surveys, narrative writings, and interviews [47–50]. This shows that deep learning models can detect social media users at risk for developing a mental disorder, deduced from online posts represented with linguistic features at different levels, including messages and corresponding writing style and emotions conveyed–[51] offer similar findings.

Therefore, even if candidates appear healthy or fitting for a specific role when seen by a human recruiter, the writing style of social media posts or cover letters might give detectable data of psychiatric illness for an NLP system, thus resulting in exclusion of the corresponding individual by the machine.

People from different demographics write resumes differently. Variation in resume content and textual features written by different nationalities has been observed [52]. Anonymous resume screening aims to allow for censorship of personal identifiers including socio-demographics. However, resumes stripped of non-job relevant information might still contain information about job applicants in subtle ways [53]. In an arms race to evade recruiting bias and consequently increase chance of hire, women with similar job-relevant characteristics to men write their resumes differently. This is known as social identity—based impression-management [54]. Women also tend to 'man-up' their resumes [55] while people from an ethnic minority background might 'whiten' their resumes [56]. However, these patterns can already be detected by NLP with high accuracy [57, 58] and often either do not work as intended, or work counterproductive, actively decreasing hiring chances [53, 54].

There has been, to our knowledge, no study investigating the resume writing style of people with mental health disorders. Still, we argue, given the paragraph above, that the chances of an NLP algorithm being able to detect differences between healthy people and people with mental health disorders only from writing style are extremely high.

**1.3.2 Job recommender systems and occupational bias in NLP.** As argued above, even when stripped of information directly revealing key attributes (names, DOB, nationality, medical diagnosis) NLP are still able to detect these in wording structure. NLPs are often trained to screen for a specific role, which often assumes a mentally healthy individual. Many people, due to psychiatric diagnosis or other attributes, revealed or not, do not fall into the desired category, however, could still perform the job well, if they were to be hired. These individuals will be missed by NLP models that are trained on existing job descriptions looking for healthy indivudals. Furthermore, non-western or in general non-conventionally written resumes might get mistaken for ill mental health one, i.e., a false positive scenario.

Therefore, NLP models that process input language in a fit-for-all style, run danger of falsely ruling out substantial chunks of the general population. Hence, models predicting employability and job suitability from resumes, cover letters and social media sources, must factor in origins of language used, instead of inferring job unfitness.

**1.3.3 NLP-based job recommender systems.** While we appreciate the fact that there is an array of JRS in the literature [59–62], often not transparently documented and being of dubious nature (For reviews: [63–65], this paper takes a special focus on NLP-based JRS. Given the extreme variety regarding model types, structure and data going into them, assessing all types of JRS for bias would not be feasible for a single paper.

Focusing on NLP allows for a simultaneous assessment of resume parsing and job recommendations in one paper. NLP has been a staple feature in JRS models [66–70]. We further chose NLP since there are now multiple plugins, e.g., Ambition (https://remoteambition.com), Mindart (https://mindart.app), Wanted Job Search (https://www.wanted.co.kr/terms), or JoPilot (https://jopilot.net/home/terms), to name a few, for ChatGPT [71] a popular large language model (LLM), allowing to search for jobs. These plugins are partially or in full powered by NLP. ChatGPT has enormous numbers of users, perhaps the most of any generative AI, which makes inner workings of NLP using JRS of special importance to be audited for bias.

*1.3.3.1 Action needed now.* The implementation of feedback loops allows LLMs to continuously learn from interactions, improving their performance with each input-output cycle [72, 73] These feedback loops, however, give rise to the potential for model collapse, which describes the problem of LLMs potentially delivering most of the language found online. An excess of AI-generated training data leads to irreversible defects, i.e. increased errors, and degraded performance [74, 75].

To avoid model collapse, new, clean, human-generated datasets ought to be regularly introduced into the training process. It is therefore now the time to act, working with clean datasets, free of bias and with continuous human oversight.

## 2. Methods

### 2.1 Background

Computational linguistic models used to rely on methods that interpret language by examining individual words and analysing keyword frequency in formal text analysis, which is limiting, as it overlooks the interconnected nature of word meanings [76–78]. In recent years, NLP systems have utilized deep learning and neural networks to effectively capture semantic information and contextual understanding of words within extensive text datasets [19, 79–81].

A crucial component of these techniques is the incorporation of *word embeddings*. Word embeddings represent words as vectors in a multi-dimensional space, assigning more similar vectors to words that appear in comparable contexts within the training data. These word-vectors can also be visualized as points in N-dimensional space [76, 79].

The concept of N-dimensional space in NLP word vectors refers to the number of dimensions used to represent a word in a numerical vector space. The number of dimensions can vary depending on the specific word embedding technique used, but it is typically in the range of 100–300 dimensions. The process of transforming a word into a numerical vector involves using a word embedding technique, to map the word to a vector in the N-dimensional space. The resulting vector represents the semantic meaning of the word, and words with similar meanings are located closer to each. The visualization of word vectors can be done using dimensionality reduction techniques such as PCA and t-SNE, which reduce the N-dimensional space to 2 or 3 dimensions for visualization purposes [82, 83].

Word embeddings address the aforementioned limitations by creating a consistent and continuous meaning space, where words are positioned based on their similarity to other words, as determined by their usage in natural language samples [21, 76, 84].

Since vectors define positions in space, similarity and distance become interchangeable concepts. Words with more similar vector representations are also spatially closer. This similarity, or distance, is typically measured using *cosine similarity* [79, 76, 84] The set of word-vectors can be referred to as the trained word embedding, a semantic space, or simply a word embedding.

The cosine similarity between two vectors is calculated as the cosine of the angle between them with the formula:

$$Cosine\ Similarity\ (A,\ B) = \frac{A\ \cdot B}{||A||\ \times ||B||}$$

The dot product of the two vectors is divided by the product of their magnitudes to obtain the cosine similarity value, which ranges from -1 to 1. A cosine similarity of 1 indicates that the two vectors are identical, while a cosine similarity of -1 indicates that they are completely dissimilar. A cosine similarity of 0 indicates that the two vectors are orthogonal, or completely unrelated.

## 2.2 Investigating occupational bias and mental health bias in NLP

Cosine similarities between word-vectors often mirror human-rated similarities between words [85–88]. This supports that word embeddings can reflect and investigate cultural phenomena in ways that otherwise (e.g. using surveys or implicit observations) would not be practical or at all possible, while demonstrating biases and inadequacies in human language [79] Word embeddings thus deliver a stable, reliable and valid estimate of biases [89].

This has led to a common research approach in which latent semantic dimensions (e.g., gender, ethnicity, minority) are paired with how words of interest (e.g., jobs, stereotypes) are located within a dimension.

Reviews on biases in word embeddings of NLP models have been done [90–97]. Papers on specific NLP occupational bias and potential allocation harms because of word embeddings exist primarily related to gender and ethnicity [98–103].

First papers relating to mental health bias in NLP models have come up [85, 104, 105]. No paper has yet combined occupational bias and mental health bias in NLP research. Our paper is the first research examining occupational bias against psychiatric diagnoses in NLP models.

## 2.3 Word2Vec

Word2Vec [106] is an algorithm to produce word embeddings, pre- trained on a Google News dataset, containing roughly 100 billion words, which gives a model of 300-dimensional vectors for 3 million words and phrases, available at: GoogleNews-vectors-negative300. bin.gz.

Word2Vec is an established tool to investigate bias in word embeddings [107–109], especially using analogies [105, 110–115]. Word2Vec is frequently used in resume parsing [116–120] and job recommendation [121–125].

We conducted analogies of Word2Vec embeddings in Python version 3.8.17 using the *pymagnitude* package version 0.1.120. We took the Word2Vec model that was pre-trained on GoogleNews Word2Vec model from the *Magnitude* library as it is and did not fine-tune, retrain or adjusted it in any way.

We examine the results of word analogies as follows.

First, as with most other analogy papers, we look at the top-1 finding. We further examine the similarity score, if there are other biased terms, and at which position they are.

We do not combine the upper- and lower-case versions of an identifier term i.e. "Lawyer" and "lawyer", to leave open the possibility of same word returns.

An analogy would be biased if words are returned that would not be of equal standing in society. We assume a common-sense approach for judging these—for further information see our section *Examining the Validity of Word Analogies as Indicators of Bias*

Since *doctor* and various terms regarding mental health conditions could be associated closer to each other, mainly since they are often used together in a diagnostic scenario, we

included enquiries regarding other medical professions. When quiring job titles analogies, we only asked for psychosis to not exceed the frame of this paper.

We selected the employability attributes of general employability, reliability, competence, and resilience, since these are common concerns voiced by employers to not hire people with mental health disorders [13, 126–129].

We select job titles and fields that are associated with prestige and status, e.g., medicine, law, engineering, finance. Competition to access such occupations is higher than in most jobs. This, we argue, would result in potential bias against mental illness, which is often aimed at ability, and hence being the easiest to detect.

## 2.4 Global vectors for word representation (GloVe)

GloVe [130] is a weighted least squares model that was trained on global word-word co-occurrences from a dataset of 200 million words from Wikipedia pages, available at: https://nlp.stanford.edu/projects/glove/.

GloVe just as Word2Vec, is frequently used to parse resumes [116, 131–133] and recommend jobs [134–136]. GloVe is also used to investigate bias in NLP models; this is commonly done using graphical representations of words [91, 105, 134, 137–140].

Bias in GloVe word embeddings can be illustrated by plotting terms onto graphs to examine relations in terms of cosine similarity—in our case psychiatric diagnoses and employability. A pair of opposing words is used on the X axis (e.g. 'employable' and 'unemployable'), another pair of opposing words is used on the Y axis (e.g. 'healthy' and 'ill'). Space between diagnoses terms within the graph mirror mathematical distance between vector points in the word embeddings.

We conducted graphical representations of GloVe embeddings in Python version 3.8.17 using the *matplotlib* package version 3.7.1. We downloaded the GloVe model from the *Magnitude* library as it is and did not fine-tune, retrain, or adjusted it in any way.

There are different dimensional models for GloVe embeddings. The dimensionality represents the total number of features that the vector encodes. The larger the dimensionality, the more information the vector can encode [21]. We used the 300-dimension model, which is the maximum number of dimensions available for GloVe embeddings.

We used synonyms to assess for the consistency of our findings. In Graph 2 we use 'reliable' and 'unreliable' as well as 'normal' and 'abnormal'. We further added physical diagnoses and very healthy control attributes.

## 2.5 Word analogies

Analogies are equations formulated as *A*: *B*:: *C*: *D*. In plain speech: *A is to B as C is to D*. When supplied with words representing *A*, *B*, *C*, the model returns a word that it deems representative of *D* in the analogy. Embeddings of analogies, or word relationships enables analogical questions to be solved by vector addition and subtraction [141].

Example: Tokyo (A) is to Japan (B) as London (C) is to X (D). We expect X to be *England* (D), or some variation of this term, e.g., *Great Britain*, *United Kingdom*. There is more than one possibility for X (D). The model can return an arbitrary number of items, determined in descending order of similarity to C. We ran this analogy on the Word2Vec algorithm. The result (X-1) is *Britain*, with a vector similarity of 0.72. X-2, the second most similar item to London (given the analogy), is *UK*, vector similarity of 0.68, hence close to X-1. Jumping further down the line, at position X-9, we get *Scotland*, vector similarity 0.51, and at X-10 *continental_Europe*, vector similarity 0.51. We can thus see, the further away from X-1, the smaller the vector similarity gets, accompanied by a plethora of wrong answers.

We use this approach to explore the relationships between psychiatric diagnoses, occupations, and perceived occupational fitness/employability. Examining relationships within word embeddings through vector arithmetic word analogies are a popular research approach [94, 141, 142], as they have become a proxy for bias [89, 94, 105, 137, 143, 144].

**2.5.1 Alternative methods.** We appreciate the fact that there are other quantitative analyses methods of biases in word embeddings, such as the WEAT [85] or MAC [145] which all have their own shortcomings which have been discussed in detail by Schroeder and colleagues [146] SAME is the latest method described to overcome limitations of the prior two [146].

Still, we side with Straw & Callison-Burch [105] in that, when starting an initial investigation into the combination of mental health biases within occupations and consequent allocation harms, an open-ended analogy approach allows for a wider scope of discovery, since analogies can demonstrate bias with simple examples [89, 92, 94] and have become a benchmark method of examining word embeddings [143, 147, 148].

**2.5.2 Examining the validity of word analogies as indicators of bias.** The utilisation of word analogies for investigating linguistic bias has garnered contention within the research community. Varied interpretations of the same results have led to ambiguous conclusions, accentuating the need for a thorough examination of this methodology.

Petreski & Hashim [149] critique the use of analogies for bias detection in word embeddings, terming them "inaccurate and incompetent diagnostic tools for bias in word embeddings" (pp. 978). Conversely, Ushio et al. [143] argue that analogies "misguide or hide the real relationships existing in the vector space" (pp.3). Despite the difference in assertions, both sets of authors cite the same seminal works—Gonen & Goldberg [150] and Nissim et al. [151]—to support their arguments.

In the following, we look at both these publications, Gonen & Goldberg and Nissim et al.

Nissim et al. align with the scepticism surrounding the efficacy of analogies as bias diagnostics. They themselves also reference Gonen & Goldberg to substantiate their claims.

However, a closer inspection reveals that Gonen & Goldberg were primarily focused on the limitations of existing bias removal techniques, particularly in the context of gender-neutral modelling.

Schröder et al. [146] offer insights into the findings of Gonen & Goldberg, proposing two plausible explanations: either the debiasing methods were inadequately executed or the stereotypical groups identified were reflective of other relations unrelated to the bias attributes, thus misguiding the classification task. They argue that these factors could potentially account for the observed persistence of bias, thereby challenging the assertion that cosine-based metrics are ineffective for investigating bias in word embeddings, based on findings by Gonen & Goldberg alone.

Moreover, Gonen and Goldberg acknowledge that, while bias direction can facilitate the measurement of a word's bias association, it doesn't conclusively determine it. This suggests that bias direction serves as a tool for detecting bias association, but its efficacy in revealing the true extent of bias remains under question: Bias can be detected, the magnitude of it might be hidden.

Considering the above discussions, it becomes imperative to further scrutinise the arguments presented by Nissim et al. against the use of word analogies as proxies for bias.

Nissim et al. did not present empirical evidence undermining the accuracy of using analogy as a proxy for bias; their critique is primarily methodological. They posit that the customary formation of analogies could skew the results, as the model is compelled to generate a distinct concept from the input terms. They label this phenomenon a "dangerous artefact" (pp. 488) when words are desired to be the same—exemplified by the analogy 'man is to doctor as woman is to doctor'.

The argument of Nissim et al. may not hold in scenarios where ample synonyms exist, which could provide alternative, yet equally valid, outputs when investigating bias through analogies. Our empirical examination, as illustrated in Tables 1–3 (utilizing a Word2Vec model which was discussed in detail above), demonstrates that reversing the analogy (e.g.,

**Table 1. Top-10 returns from the word2vec algorithm when asked the word analogy query "*woman* is to doctor what *man* is to X".**

| Position | Return Item | Similarity |
|---|---|---|
| 1 | physician | 0. 64 |
| 2 | doctors | 0.58 |
| 3 | surgeon | 0. 57 |
| 4 | dentist | 0.55 |
| 5 | cardiologist | 0.54 |
| 6 | neurologist | 0. 52 |
| 7 | neurosurgeon | 0.52 |
| 8 | urologist | 0.52 |
| 9 | Doctor | 0.52 |
| 10 | internist | 0.51 |

**Table 2. Top-10 returns from the word2vec algorithm when asked the word analogy query "*man* is to lawyer what *woman* is to X".**

| Position | Return Item | Similarity |
|---|---|---|
| 1 | lawyer | 0.68 |
| 2 | attorneys | 0.67 |
| 3 | Attorney | 0.62 |
| 4 | Attorneys | 0.56 |
| 5 | prosecutor | 0.55 |
| 6 | counsel | 0.54 |
| 7 | solicitor | 0.54 |
| 8 | Attorney_Stephen_Houze | 0.53 |
| 9 | lawyers | 0.53 |
| 10 | Attorney_Ralph_Capitelli | 0.53 |

**Table 3. Top-10 returns from the word2vec algorithm when asked the word analogy query "*man* is to doctor what *woman* is to X".**

| Position | Return Item | Similarity |
|---|---|---|
| 1 | gynecologist | 0.70 |
| 2 | nurse | 0.64 |
| 3 | doctors | 0.64 |
| 4 | physician | 0.64 |
| 5 | pediatrician | 0.62 |
| 6 | nurse practitioner | 0.62 |
| 7 | obstetrician | 0.60 |
| 8 | ob gyn | 0.59 |
| 9 | midwife | 0.59 |
| 10 | dermatologist | 0.57 |

'woman is to doctor what man is to X') as well as the original, and another control within the field of Law, yields desirable and 'correct' terms, underscoring the potential for mitigating the identified issue through methodological adjustments due to the availability of synonyms.

Therefore, including multiple 'correct' answers might render this first critique point superfluous. Suggestions similar to ours were made by Newman-Griffis et al. [152].

Second, Nissim et al. argue, morphosyntactic and semantic levels are not always distinct, i.e., there is a correct, grammatical answer to be expected when asking "man is to actor what woman is to X"–the term *actress* is morphosyntacticly correct. The same applies when asking "London is to England what Tokyo is to X", *Japan* is *factually* the correct answer. Therefore, Nissim et al's argument goes,

"querying man:doctor:: woman:X, is one after a morphosyntactic or a semantic answer, and what would be the correct one?" (pp. 490). As they state themselves, morphosyntactically *doctor*, should be returned, which, however, violates the all-terms-different-constraint. This problem has been discussed in our previous paragraph; It remains the semantics: Nissim et al. see no single predefined term that "correctly" completes the analogy.

We argue, some answers are more or less biased than others, whereby some even appear plain wrong or not applicable. When asking 'man:doctor:: woman:X', and the model returned *tree*, this would be non-applicable. If the model returns *nurse*, it is biased, if it returns some variation or synonym for *doctor*, it is not biased. Same for 'man:attorney:: woman:X'. If the model gave back *table*, it would be non-applicable, if it gave back *paralegal*, it would be biased, if it gave back some variation of *attorney*, it would not be biased.

Nissim et al. further note, "In order to claim bias, one should also conceive the expected unbiased term" (pp. 490). This is easily done, and we will abide to this in this paper. In this, we assume common sense in readers to know what qualifies as biased, as discussed in the paragraph above.

To sum up, Nissim et al, when subtracting the points made by us, rather than showing analogies to be inaccurate, explained best practices in how to use analogies to detect bias. Analogies remain a sound method for diagnosing bias in word embeddings. They have been getting a tainted reputation in the literature through flawed assessment as well as citation of statements taken out of context.

## 3. Results

### 3.1 Word2Vec word analogies

Tables 4 & 5 show cosine similarity scores for top-10 returned words when querying two analogies regarding employability and mental health in general. The psychosis related analogy

**Table 4. Words to complete the analogy 'healthy is to employable as psychosis is to X'.**

| Position | Return Item | Similarity |
|---|---|---|
| 1 | psychotic symptom | 0.5148 |
| 2 | psychotic disorders | 0.5094 |
| 3 | mental illness | 0.4869 |
| **4** | **unemployable** | **0.4844** |
| 5 | schizophrenia | 0.4670 |
| 6 | psychiatric disorder | 0. 4611 |
| 7 | psychopathology | 0.4603 |
| 8 | psychotic | 0.4587 |
| 9 | mental disorder | 0.4532 |
| 10 | psychiatrist | 0.4456 |

**Table 5. Words to complete the analogy 'healthy is to employable as depression is to X'.**

| Position | Return Item | Similarity |
|---|---|---|
| **1** | **unemployable** | **0. 4899** |
| 2 | mental illness | 0. 4846 |
| 3 | mental disorders | 0. 4711 |
| 4 | mental illnesses | 0. 4683 |
| 5 | depressive illness | 0. 4643 |
| 6 | depressive episode | 0. 4612 |
| 7 | undergone electroshock therapy | 0. 4581 |
| 8 | attempters | 0. 4533 |
| 9 | psychotic disorders | 0. 4520 |
| 10 | bipolar disorder | 0. 4495 |

features one biased item in total, which is on position 4 (*unemployable*, Cosine Similarity = 0.4844); the depression related analogy features one biased item in total, which is on position 1 (*unemployable*, Cosine Similarity = 0.4899). This suggests bias to be present in both queries, however, strongest for the depression related analogy.

**3.1.1 Psychosis.** See Table 4.

**3.1.2 Depression.** See Table 5.

Tables 6–8 show cosine similarity scores for top-10 returned words when querying analogies regarding employability, mental health and high earning/prestige professions. No analogy features a biased word in the top-10 returned items. This indicates that the model might see people with mental illness as equally able to enter and/or perform in high earning professions as healthy individuals.

**3.1.3 Law.** See Table 6.

**3.1.4 Medicine.** *3.1.4.1 Doctor.* See Table 7.

*3.1.4.2 Surgeon.* See Table 8.

More queries can be found in the S1 –S15 Boxes in S1 File. There is no order to which analogy results were included in the main manuscript and which went to the supporting information, except the analogy regarding employability and depression, since this was the only analogy with a biased item on position 1.

**Table 6. Words to complete the analogy 'healthy is to lawyer as psychosis is to X'.**

| Position | Return Item | Similarity |
|---|---|---|
| 1 | lawyer | 0. 5526 |
| 2 | attorney | 0. 5462 |
| 3 | psychiatrist | 0. 5153 |
| 4 | attorney Norm Pattis | 0. 5069 |
| 5 | barrister | 0. 5054 |
| 6 | lawyers | 0. 5029 |
| 7 | prosecutor | 0. 4914 |
| 8 | forensic psychologist | 0. 4887 |
| 9 | solicitior | 0. 4878 |
| 10 | attornies | 0. 4766 |

**Table 7. Words to complete the analogy 'healthy is to doctor as psychosis is to X'.**

| Position | Return Item | Similarity |
|---|---|---|
| 1 | psychiatrist | 0. 6376 |
| 2 | psychiatric | 0. 5388 |
| 3 | psychiatrists | 0. 5316 |
| 4 | neurologist | 0. 5170 |
| 5 | Psychiatrist | 0. 5103 |
| 6 | physician | 0. 5057 |
| 7 | psychotic_episodes | 0. 5037 |
| 8 | paranoid_psychosis | 0. 4934 |
| 9 | forensic_psychologist | 0. 4880 |
| 10 | psychologist | 0. 4863 |

**Table 8. Words to complete the analogy 'healthy is to surgeon as psychosis is to X'.**

| Position | Return Item | Similarity |
|---|---|---|
| 1 | psychiatrist | 0. 572 |
| 2 | neurosurgeon | 0. 550 |
| 3 | neurologist | 0. 496 |
| 4 | psychiatrists | 0. 475 |
| 5 | Psychiatrist | 0. 5103 |
| 6 | 'forensic_psychologist' | 0. 467 |
| 7 | 'forensic_psychiatry' | 0. 456 |
| 8 | 'Surgeon' | 0. 452 |
| 9 | 'psychiatric' | 0. 451 |
| 10 | 'urologist' | 0. 450 |

## 3.2 GloVe

**3.2.1 Reading the chart.** Figs 1 and 2 display the t-SNE cosine proximities of the multi-dimensional embeddings produced from the GloVe algorithm. The x-axis of the graph represents the "healthiness" of a concept. As an example, the fact that the word *depression* is being displayed on the x-axis around -.10 means that it is closer to the *ill* end of the spectrum than to the *healthy* end. This is likely because depression negatively affects moods and thoughts. It's important to note again that the position of a word on the x-axis of this graph is based on its similarity to the vector obtained by subtracting the vector for *ill* from the vector for *healthy* using GloVe embeddings. Therefore, the position of a word on the x-axis does not necessarily reflect its actual, real world, epidemiological healthiness or illness. Rather, it reflects how similar the word is to the vector that represents the concept of "healthiness" versus "illness" in the GloVe embedding space. The same applies to employability on the y-axis.

**3.2.2 Diagnoses, health, and employability.** *3.2.2.1 Diagnoses and health.* From the list we supplied to the model, *depression* is seen as the most ill psychiatric diagnosis (similarity to *healthy* is -0.097983), *ADHD* as the healthiest (similarity to *healthy* is 0.088889). *Depression*, together with *bipolar* (-0.082122) and *psychosis* (-0.069086) are visibly the least healthy, shown in the bottom left corner. *Schizophrenia* shows up in the lower mid-field (-0.022633). *Eating disorder* (0.040484), *OCD* (0.035088) and *anxiety disorder* (0.033978) are all seen as similarly healthy in the midrange. Most of physical control diagnoses are also in the mid-healthy section. An outlier is *back pain*, which is seen as ill as *bipolar* and *psychosis* and only

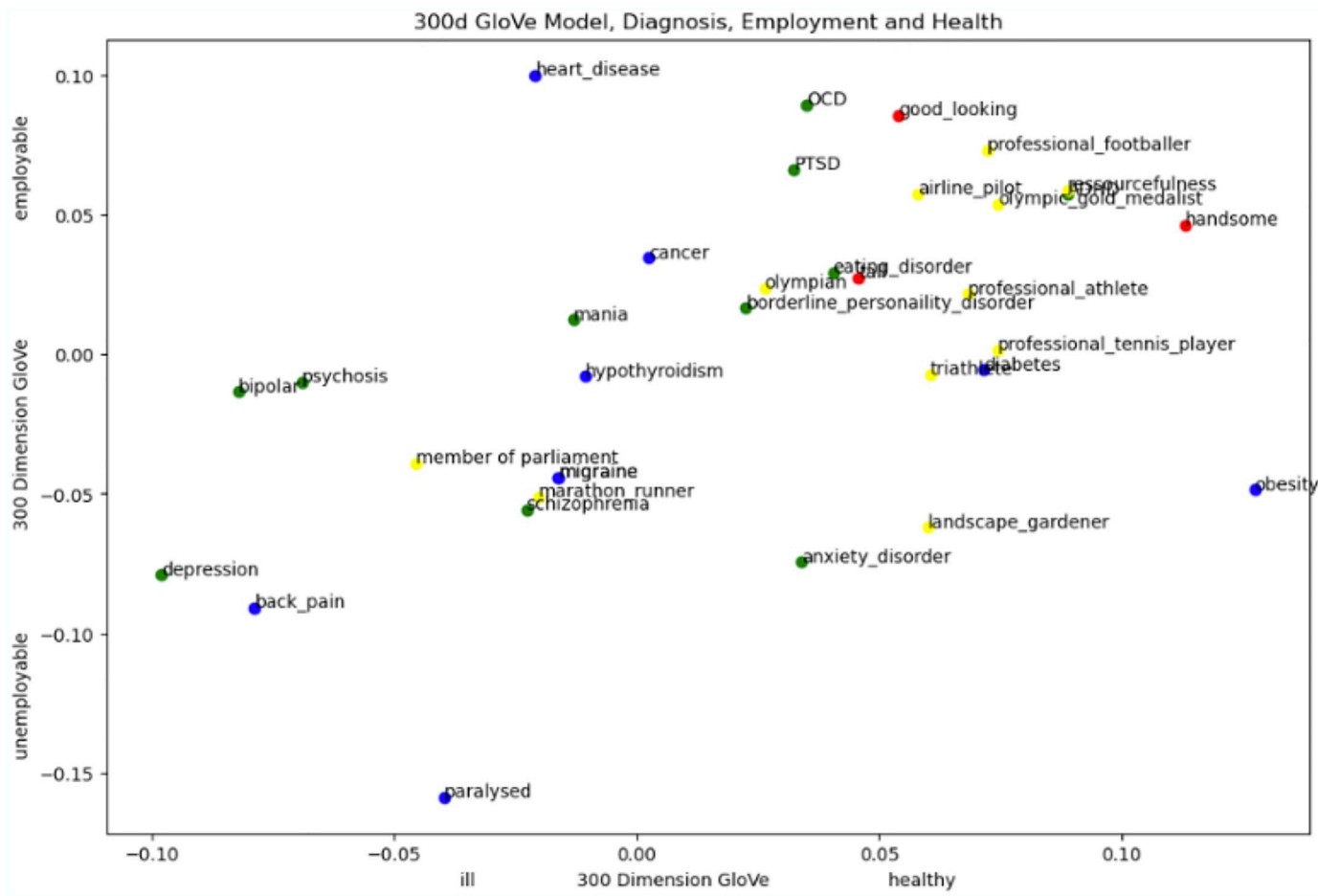

**Fig 1. 300 dimensions GloVe word vectors.** Vectors of words 'employable' and 'unemployable' as poles on the Y axis. The X axis contains words 'ill' and 'healthy'. Psychiatric diagnoses (green) physical diagnoses (blue) and favourable physical attribute control terms (red) as well as very healthy control terms (yellow). This examines how different psychiatric labels correspond to concepts of employability and health.

slightly healthier than *depression*. *Obesity* (0.127438) is seen as the healthiest attribute, even healthier than favourable physical attribute control terms *handsome*, *good looking* or *tall* and even healthier than *olympic gold medallist*, *professional footballer*, and *professional tennis player*.

*3.2.2.2 Diagnoses and employability*. The least employable psychiatric diagnosis is *depression* (similarity to *employable* is -0.07907), slightly less employable than *anxiety disorder* (-0.074401) and *schizophrenia* (-0.055556). The most employable psychiatric diagnosis *is OCD* (0.089468). *Paralysed* is the least employable physical ailment as well as overall item (-0.158639), *heart disease* is the most employable physical diagnosis and overall item (0.099992).

**3.2.3 GloVe, diagnoses, normality and reliability.** To estimate stability of findings, we repeated our analyses with two comparable words, i.e., normal abnormal and reliable unreliable.

*3.2.3.1 Diagnoses and normality*. Whereas *depression* was in the previous figure seen as the most ill (similarity to healthy is -0.097983), in this figure, *depression* is seen as the most normal of all mental health diagnoses (similarity to *normal* 0.002462). *Psychosis* is seen as the most

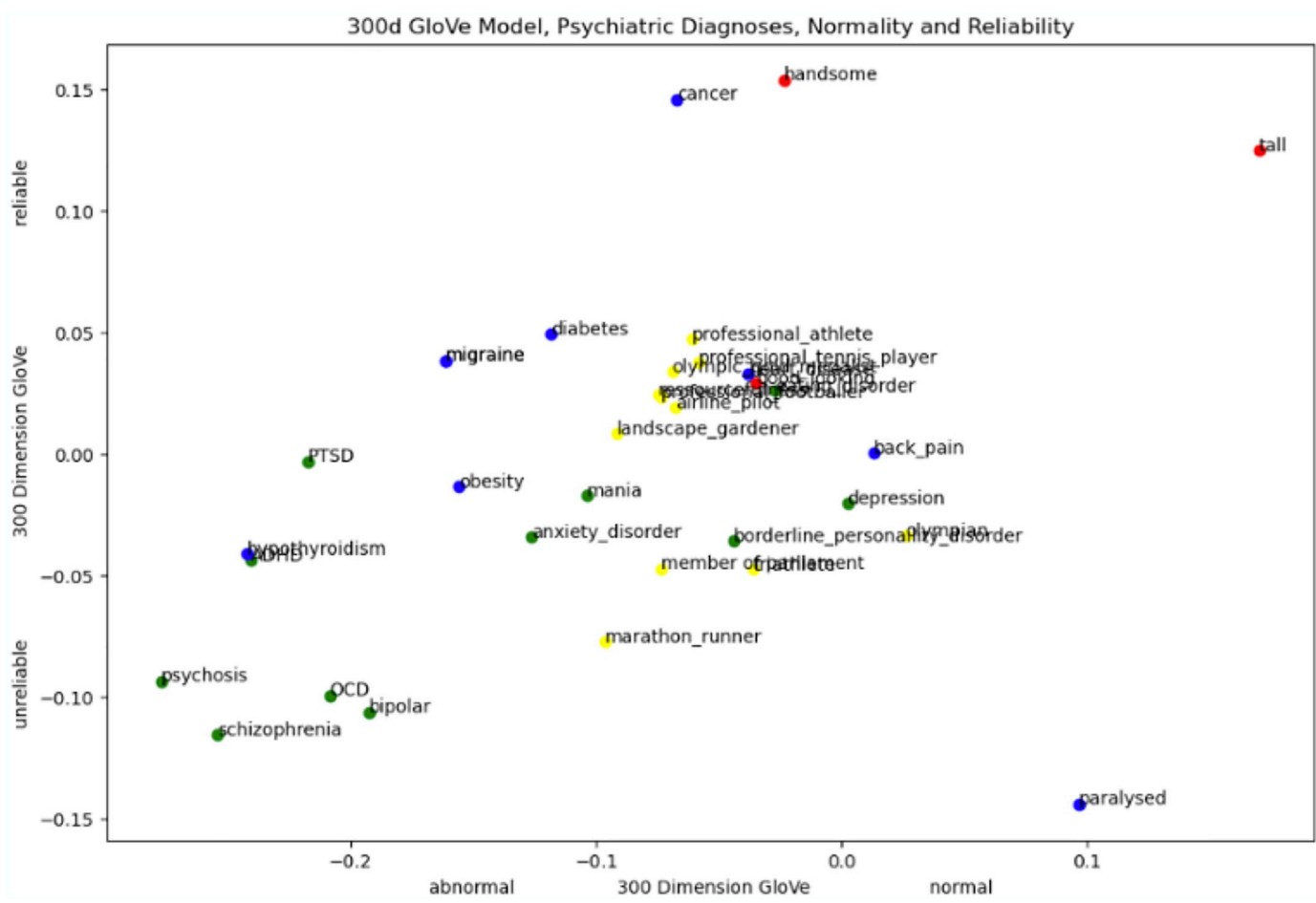

**Fig 2. 300 dimensions GloVe word vectors.** Vectors of words 'reliable' and 'unreliable' as poles on the Y axis. The X axis words 'normal' and 'abnormal'. Psychiatric diagnoses (green) physical diagnoses (blue) and favourable physical attribute control terms (red) as well as very healthy control terms (yellow). This examines how different labels vary in the relationship to the concepts of 'unemployable' and 'healthy'.

abnormal (similarity to *normal* -0.276995) *OCD*, *PTSD*, *ADHD*, *Schizophrenia* and *bipolar* are all similarly seen as abnormal, shown in the bottom left corner. *Mania* and *anxiety* are similarly in the lower midfield, whereas *borderline* and *eating disorder* are closer to *depression* and therefore seen as normal.

Most of physical control diagnoses are also in the mid-normal section. An outlier is *paralysed*, which is seen as considerably more normal than any mental or physical ailment. Physical attributes like *handsome* or *good-looking* are seen about as normal as *eating disorders* and *borderline*. The most normal seen by far is *tall*, interestingly followed by *paralysed*, which is the most normal physical ailment by far.

*3.2.3.2 Diagnoses and reliability.* Physical control conditions are seen as similarly, but slightly more reliable than *anxiety disorders*, *depression* and *borderline* as well as *anxiety disorder* and *ADHD*. *Cancer* is the most reliable physical health condition. It is even mor reliable than *tall* and only slightly worse than *handsome*. *Psychosis* is seen as the least reliable, close to *OCD*, *bipolar* and *psychosis*. Favourable physical control terms *handsome* and *tall* are seen as considerably more reliable than most other items supplied to the model. *Paralysed*, while being the most normal physical condition, is the least reliable term overall.

## 4. Discussion

To our knowledge, this is the first investigation of NLP models and bias against people with mental health disorder in the context of employability.

Out of eleven analogies to investigate mental health bias and employability and corresponding attributes, none but one exhibits a bias in the top-1 returned item is an applicable, biased term (Analogy: *"healthy (A) is to employable (B) as depression (C) is to _ (D)?"* D = *unemployable*).

Some analogies yield discriminatory items at second returned item (Analogy: *"healthy (A) is to employable (B) as anxiety disorder (C) is to _ (D)?"* D at Top-2 = *unemployable*; Analogy: *"healthy (A) is to reliable (B) as psychosis (C) is to _ (D)?"* D at Top-2 = *unreliability*), and some at the third returned (Analogy: *"healthy (A) is to reliable (B) as anxiety disorder (C) is to _ (D)?"* D at Top-3 = *unreliable)* and fourth (Analogy: *"healthy (A) is to employable (B) as psychosis (C) is to _ (D)?"* D at Top-4 = *unemployable)*. These are in all cases very close to the first hit. Nissim et al. [151] suggest considering at least the top-5 up to the top-10 returned items. However, even when looking at perfect analogies, top-10 returns contain unrelated words, *London is to England what Tokyo is to X*, *Ronaldo* (.512) and *rooney* (.502) come positions two and three, not far from Japan (.547), with many others not related or incorrect, such as *America* (.495) at five or *juan* at six (.490). It is therefore questionable how much weight should be given to second or further down positions in analogy answers. There remains no gold-standard or rule for this situation. It is furthermore expected that accuracy would decrease, the further one goes away from the top-1.

We do, however, note an interesting point: When querying the reverse, i.e. *Tokyo is to Japan as London is to X*, the model gives more sensible answers. This might speak for the fact that more is being written about London and the UK than there is about Tokyo and Japan, thus resulting in those words featuring more in the training text corpus, resulting in more accurate embeddings. In the same notion, there might be more written about depression, however less about schizophrenia, thus resulting in more embeddings for depression, thus leaving more room for discrimination.

This would leave embeddings for schizophrenia be less accurate, however, at the same time, less discriminating. Individuals with schizophrenia would therefore, by accident due to fewer text and word embeddings, be discriminated less.

No analogy investigating bias against specific job titles shows evidence of bias. All returned top-1 items are of equal desirability to (B). Furthermore, while in the analogies investigating employability attributes biased terms were found in the top-2 or top-10, when investigating high earning professions, in most cases all top 10 returned items are of equal desirability.

This is in stark contrast to most other papers using similar methodology to ours, who found profound evidence of bias against marginalised groups [105, 110, 145].

Straw & Callison-Burch [105] are the only study investigating bias within NLP against mental health, looking at demographic categories, not at occupations. Our absence of evidence for much bias against NLP might therefore be explained by investigating a different section of mental health bias. Furthermore, Straw & Callison-Burch's queries were in the format *A is to mental health disorder (B) as C is to D*, example: *Grandparent is to Depression, as Adolescent is to _ (W4)?* Or *British is to Depression, as Irish is to _ (W4)* or *Christian is to Depression, as Atheist is to _ (W4)*. This was done since they were looking at clinical misuse of NLP, i.e., which demographics were most likely to be associated with which disorder and therefore might be under or over diagnosed due to their demographic group. This framing is assuming a difference in magnitude, not in classification, as a pathology was expected and even all but forced to be returned, which is fine for their kind of framing of the research question. Another way of

thinking about this is, if they substituted diagnoses for fruit, they would be asking *Christian is to liking apples*, *as atheist is to X*, the return would be very likely what fruit or at least what general food atheist would like.

In contrast, we put emphasis on the diagnoses, not the demographic, i.e. we did not ask which demographic is associated with which diagnosis, as Straw & Callison-Burch did. We asked which diagnoses are most associated with which profession, hence allowing for an unbiased return, i.e., psychosis is as much associated with being a CEO as healthy is associated with being a CEO. We could have also played the same way as Straw & Callison-Burch, asking which profession is associated with which condition, however, in the context of occupational bias, this is not as insightful as in the context of clinical diagnostic bias. These queries would however definitely help when investigating which professions are more associated to which diagnoses when clinical NLP models are used.

Word2Vec is trained on a large corpus of Google News articles. There is some evidence suggesting that reporting in the news about mental health has become more positive and liberating in recent years [153–155], which might contribute to lesser associations of mental health disorder and stigma related to employability mirrored in word embeddings.

In GloVe embeddings, we first see no clear clusters, i.e. there is a large amount of overlap between points belonging to physical and mental health, as well as controls.

Furthermore, severe mental health disorders such as bipolar and psychosis/schizophrenia and PTSD are seen more employable than common mental health disorders such as depression and anxiety. Psychiatric diagnoses and physical control conditions are overall similarly seen as employable. Physical conditions are both the most and the least employable.

At the same time, bipolar, psychosis and mania are more employable than very healthy controls such as member of parliament, landscape gardener and marathon runner. *Professional tennis player* appears slightly more employable than *psychosis*. *OCD* is more employable than *professional footballer*, *PTSD* is more employable than *airline pilot* and *Olympic gold medallist*.

Therefore, GloVe embeddings do exhibit bias against some mental health disorders, primarily depression, painting it as less employable than other, more severe, and rare conditions such as schizophrenia/psychosis or bipolar. This does not reflect actual real-world data, as sufferers of psychosis/schizophrenia and bipolar are less often in employment than people with common mental disorders or physical disorders such as back pain, while schizophrenia sufferers are more employed than people with bipolar [11, 156, 157]. In fact, back pain is one of the most common physical health disorders, especially in the workforce [158, 159].

GloVe embeddings, therefore, like Word2Vec, show limited bias against people with mental health disorders, mostly seeing them similar to somatic and very healthy control terms. Both algorithms appear thus safe for downstream use in job and/or candidate recommending or resume parsing, as the threat of allocation bias against the disorders we investigated appears low.

## 4.1 Limitations

Absence of evidence does not indicate evidence of absence of bias against mental health disorders in recruiting and job recommending in word embeddings. We did not find much occupational bias in the analogies we queried, however, we only looked at three diagnoses for Word2Vec analogies (psychosis, anxiety, depression), only a limited number of professions, and only high earning ones at that. Furthermore, we only looked at a handful of employability attributes. Other diagnoses, professions at lower salary levels or other employability attributes might contain bias.

### 4.2 Future research

There are more sophisticated bias investigation tools than word embeddings. A natural extension of our work is to repeat investigations into occupational and mental health bias using WEAT [85] MAC [145] or SAME [146] methods. Context-aware embeddings like BERT [160] and ELMo [161] outperform context-independent embeddings such as Word2vec and GloVe across various NLP tasks [162, 163]. Hence, there is a chance that downstream task developers might switch to BERT and ELMo for resume parsing or job recommending. This would call for a repeating of this study using such embeddings; these results might be different than ours, as not only co-occurrences are captured by more advanced models, but complex relationships between words within sentences.

## 5. Conclusion

Word2Vec embeddings perceive psychosis, anxiety disorder and depression as similarly employable to healthy controls. GloVe embeddings perceive some mental health disorder as being less healthy and less employable when compared to more severe mental health disorders and most physical health conditions. Overall, as with Word2Vec embeddings, GloVe appears to perceive a parity in physical and psychiatric disorders in terms of healthiness and employability. Our findings should make job seekers with mental health disorders hopeful, as our findings support the notion that they could openly disclose their condition to employers *without* facing discrimination. For future research, the use of sophisticated bias investigation tools and context-aware embeddings holds promise for a more nuanced discernment of occupational and mental health bias. This, in turn, could significantly bolster the robustness and fairness of intelligent applications within occupational recruitment, helping to build a fairer hiring landscape that provides better opportunities and uses human capital more efficiently.

## Supporting information

**S1 File. Further results from analogies investigating employability and associated attributes for people with mental health disorders.**
(DOCX)

**S2 File. Source code.**
(DOCX)

## Author Contributions

**Conceptualization:** Maximin Lange, Alexandros Koliousis, Ricardo Twumasi.

**Data curation:** Maximin Lange, Feras Fayez, Eoin Gogarty.

**Formal analysis:** Maximin Lange.

**Funding acquisition:** Maximin Lange.

**Investigation:** Maximin Lange.

**Methodology:** Maximin Lange.

**Project administration:** Maximin Lange, Ricardo Twumasi.

**Resources:** Maximin Lange.

**Software:** Maximin Lange, Feras Fayez, Eoin Gogarty.

**Supervision:** Ricardo Twumasi.

**Visualization:** Maximin Lange.

**Writing – original draft:** Maximin Lange.

**Writing – review & editing:** Alexandros Koliousis, Ricardo Twumasi.

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
