## [Decision Letter · Decision Letter 0]

1 Mar 2024

PONE-D-24-00286Schizophrenia More Employable Than Depression? Language-Based Artificial Intelligence Model Ratings for Employability of Psychiatric Diagnoses and Physical and Healthy ControlsPLOS ONE

Dear Dr. Lange,

Thank you for submitting your manuscript to PLOS ONE. After careful consideration, we feel that it has merit but does not fully meet PLOS ONE’s publication criteria as it currently stands. Therefore, we invite you to submit a revised version of the manuscript that addresses the points raised during the review process.

We look forward to receiving your revised manuscript.

Kind regards,

Jayesh Soni

Academic Editor

PLOS ONE

Journal Requirements:

"This project was funded by the London Interdisciplinary Social Science Doctoral Training Programme (LISS-DTP)"

"N/A"

4. In the online submission form, you indicated that your data will be submitted to a repository upon acceptance.  We strongly recommend all authors deposit their data before acceptance, as the process can be lengthy and hold up publication timelines. Please note that, though access restrictions are acceptable now, your entire minimal  dataset will need to be made freely accessible if your manuscript is accepted for publication. This policy applies to all data except where public deposition would breach compliance with the protocol approved by your research ethics board. If you are unable to adhere to our open data policy, please kindly revise your statement to explain your reasoning and we will seek the editor's input on an exemption. 

5. Please ensure that you refer to Figures 1 and 2 in your text as, if accepted, production will need this reference to link the reader to the figure.

6. We note you have included a table to which you do not refer in the text of your manuscript. Please ensure that you refer to All Tables in your text; if accepted, production will need this reference to link the reader to the Table.

Reviewers' comments:

Reviewer's Responses to Questions

**Comments to the Author**

1. Is the manuscript technically sound, and do the data support the conclusions?

Reviewer #1: Partly

Reviewer #2: Yes

2. Has the statistical analysis been performed appropriately and rigorously? 

Reviewer #1: Yes

Reviewer #2: Yes

3. Have the authors made all data underlying the findings in their manuscript fully available?

Reviewer #1: Yes

Reviewer #2: Yes

4. Is the manuscript presented in an intelligible fashion and written in standard English?

Reviewer #1: Yes

Reviewer #2: Yes

5. Review Comments to the Author

Reviewer #1: This paper investigates whether Schizophrenia is More Employable Than Depression. Language Based Artificial Intelligence Model Ratings for Employability of Psychiatric Diagnoses and Physical and Healthy Controls. Thank you for this interesting read. The paper focuses on a pertinent topic and utilises a novel approach, but could be improved with a more systematic approach to testing. I have provided some comments below which I hope will help with improving the manuscript.

Title:

The title describes the aim of the paper well and is an appropriate length for the journal.

Abstract:

Clearly written and stays focused. The conclusion is maybe too definitive, when there are some clear limitations to the findings noted.

Introduction:

The introduction is interesting and covers an increasingly important topic. I think it is lacking in some focus around the key topic of employability in psychiatric diagnoses. Suggest including more information around this, perhaps some statistics showing objectively that this is the case.

Very interesting points around how language in CVs can show differences when written by different demographics. Some further examples of this, and some examples of how this has bene shown to be the case in psychiatric conditions would be beneficial for the paper.

Quite casually written in places – could perhaps be restructured to a more academic style. E.g. the sentence “Similarly, on the other end of the job searching stick, job openings had to be looked up manually by individuals in the market for a new position. Especially online, options could be endless.”.

Sentence structure can be difficult to read where sentences are particularly lengthy – section 1.2, paragraph 3 has 5 commas. Consider splitting up to multiple sentences in areas like this.

Methods:

Methods well described. Plenty of detail included and easy to follow for the reader.

t-SNE is not explained. Please provide this in core text.

Results:

Results are exhaustive and can be difficult to follow in places. Suggest improving presentation of this section – perhaps using tables or similar to present the analogies and vector results more succinctly. Could some be migrated to supplementary information so only the core questions are included?

The graphs are presented very well and a strong inclusion for the section.

Discussion:

More emphasis needed around comment that these results are not in agreement with other research in this area. Why is this the case?

Notably, the authors state that vector results past 1 may not be accurate as is seen in the test analogy London is to England as Tokyo is to ____. The vector scores in this instance are quite close to the top scoring response. Should these be written off across the board or perhaps the method isn’t functioning as expected and a different approach to measure this should be taken?

Without context on how writing in CVs of those with psychiatric disorders may differ from those without, it’s difficult to judge whether this is the more appropriate approach to analysing this. It doesn’t seem as though it can be definitively stated that certain psychiatric conditions are viewed less favourably than physical conditions. The analogy method, though a useful exercise in building the foundations of how conditions are viewed relative to healthy controls, a more systematic and informed approach is possibly needed to unravel this question.

Minor comments:

Last sentence in final paragraph of 1.1 Background has two full stops.

Second last paragraph in section 2.4. is missing an apostrophe.

Second last line of section 3.2.2. there are some font size inconsistencies.

Second line of second paragraph in section 4.2. there is highlighted text where there shouldn’t be.

Reviewer #2: This is a very interesting paper, especially because it brings to light a widely used set of AI tools aiming specifically at the bias potential that such tools may have, with focus on mental health stigma in recruiting tasks. It is very interesting to realize that mental illness of different types as well as personal features are seen as more or less reliable, normal or employable using such tools.

I read this paper very carefully, because I am far from being an AI connaisseur, but very interested in linguistics and its importance in mental health, and from my perspective, it is interestingly written, thorougly conducted and, for that reason, I find it suitable for publication.

6. PLOS authors have the option to publish the peer review history of their article (what does this mean?). If published, this will include your full peer review and any attached files.

Reviewer #1: No

Reviewer #2: No

---

## [Author Response · Author response to Decision Letter 0]

2 Apr 2024

Rebuttal Letter

Reviewer: The conclusion is maybe too definitive, when there are some clear limitations to the findings noted.

Answer: Abstract has been amended accordingly.

Reviewer: The introduction is interesting and covers an increasingly important topic. I think it is lacking in some focus around the key topic of employability in psychiatric diagnoses. Suggest including more information around this, perhaps some statistics showing objectively that this is the case.

Answer: We appreciate this comment and have amended this section accordingly, having added more comprehensive research. 

Reviewer: Very interesting points around how language in CVs can show differences when written by different demographics. Some further examples of this (…)

Answer: Further examples have been included in the addressed section

Reviewer: (continued from last query) and some examples of how this has been shown to be the case in psychiatric conditions would be beneficial for the paper. 

Answer: Thank you for pointing out this important distinction, which we didn’t make clear in the paper. We did not find any investigations into writing style of people with mental health disorders (this would be an excellent idea for a future paper). What we argue, instead is the following: Specific word use patterns of people with mental illness have been identified, while detection of general mental illnesses and even reliable estimation of population mental health is possible through analyses of social media posts, surveys, narrative writings, and interviews. At the same time, deep learning models can detect social media users at risk for developing a mental disorder, deduced from online posts represented with linguistic features at different levels, including messages and corresponding writing style and emotions conveyed. Hence, even though no study has looked at resume writing by people with mental health disorders, we argue, the chances of an NLP algorithm being able to detect differences between healthy people and people with mental health disorders would be extremely high, especially with the current speed of change within NLP techniques. We have rewritten the relevant paragraph in the paper to address this point.

Reviewer: Quite casually written in places – could perhaps be restructured to a more academic style. E.g. the sentence “Similarly, on the other end of the job searching stick, job openings had to be looked up manually by individuals in the market for a new position. Especially online, options could be endless.”.

Answer: Amended

Reviewer Sentence structure can be difficult to read where sentences are particularly lengthy – section 1.2, paragraph 3 has 5 commas. Consider splitting up to multiple sentences in areas like this. 

Answer: Amended

Reviewer: t-SNE is not explained. Please provide this in core text. 

Answer: t-SNE is already explained in section 2.1, third paragraph. However, if the reviewer wishes further explanation we will add it. We further added a short sentence to the beginning of section 3.2.1.1.

Reviewer: Results are exhaustive and can be difficult to follow in places. Suggest improving presentation of this section – perhaps using tables or similar to present the analogies and vector results more succinctly. Could some be migrated to supplementary information so only the core questions are included?

Answer: Fair suggestion – we did migrate most of the findings to the supplementary material and only left core questions included.

Reviewer: More emphasis needed around comment that these results are not in agreement with other research in this area. Why is this the case?

Answer: We believe we have already offered sufficient explanation for this in the paragraphs six to eight in the discussion section of the paper. We dedicated two paragraphs to the differences in our methodology (e.g., Straw and Callison-Burch (2020) forced a biased return) and one paragraph to our hypothesis that differences might be due to reporting in the news about mental health has become more positive and liberating in recent years, which might contribute to lesser associations of mental health disorder and stigma related to employability mirrored in word embeddings. 

Reviewer: Notably, the authors state that vector results past 1 may not be accurate as is seen in the test analogy London is to England as Tokyo is to ____. The vector scores in this instance are quite close to the top scoring response. Should these be written off across the board or perhaps the method isn’t functioning as expected and a different approach to measure this should be taken? Without context on how writing in CVs of those with psychiatric disorders may differ from those without, it’s difficult to judge whether this is the more appropriate approach to analysing this. It doesn’t seem as though it can be definitively stated that certain psychiatric conditions are viewed less favourably than physical conditions. The analogy method, though a useful exercise in building the foundations of how conditions are viewed relative to healthy controls, a more systematic and informed approach is possibly needed to unravel this question. 

Answer: We use the same method, perhaps even better executed, as Straw and Callison-Burch (2020; since our model is not forcing a biased return). Claiming the method isn’t functioning as expected and a different approach to measure this should be taken, would render Straw & Callison-Burch’s paper redundant as well. 

There remains no gold-standard or rule for evaluating analogy returns. Nissim et al., (2020) suggest considering top-5 up and sometimes the top-10 returned items. It is furthermore expected that accuracy would decrease, the further one goes away from the top- 1. We do, however, note an interesting point: When querying the reverse, i.e. Tokyo is to Japan as London is to X, the model gives more sensible answers. 

This might speak for the fact that more is being written about London and the UK (in English) than there is about Tokyo and Japan (the word2vec we investigate was trained on Google News articles) thus resulting in those words featuring more in the training text corpus, resulting in more accurate embeddings. In the same notion, there might be more written about depression, however less about schizophrenia, thus resulting in more embeddings for depression, thus leaving more room for discrimination as they are being picked up more frequently. 

This would leave embeddings for schizophrenia be less accurate, however, at the same time, less discriminating. Individuals with schizophrenia would therefore, by accident due to fewer text and resulting word embeddings, be discriminated less.

Above we already argued for the fact that we could not find in the literature any investigations into writing style of people with mental health disorders. Still, specific word use patterns of people with mental illness have been identified. Hence, even though no study has looked at resume writing by people with mental health disorders, we argue, given the evidence presented in the earlier paragraph, the chances of an NLP algorithm being able to detect differences between healthy people and people with mental health disorders would be high. 

Of course, as we acknowledge in Section 2.5.1 as well as 4.2, there are more sophisticated measures than word analogies, but, like we argue in the mentioned section, we still believe in validity and relevance of our methodology.

Reviewer : Minor comments:

Last sentence in final paragraph of 1.1 Background has two full stops.

Second last paragraph in section 2.4. is missing an apostrophe.

Second last line of section 3.2.2. there are some font size inconsistencies.

Second line of second paragraph in section 4.2. there is highlighted text where there shouldn’t be. 

Answer: All have been amended.

---

## [Decision Letter · Decision Letter 1]

9 May 2024

PONE-D-24-00286R1­­­­Schizophrenia more employable than depression?

Language-based artificial intelligence model ratings for employability of psychiatric diagnoses and somatic and healthy controlsPLOS ONE

Dear Dr. Lange,

Thank you for submitting your manuscript to PLOS ONE. After careful consideration, we feel that it has merit but does not fully meet PLOS ONE’s publication criteria as it currently stands. Therefore, we invite you to submit a revised version of the manuscript that addresses the points raised during the review process.

We look forward to receiving your revised manuscript.

Kind regards,

Jayesh Soni

Academic Editor

PLOS ONE

Journal Requirements:

Additional Editor Comments:

In the methods section, you describe the use of Word2Vec and GloVe algorithms to assess bias in NLP models concerning employability. Can you clarify if these models were retrained with any specific modifications or if they were used as pre-trained models directly? Furthermore, how did you ensure that the analogy tasks and cosine similarity measures were sufficiently sensitive and specific to detect subtle forms of bias that might not be overt?

The results section provides a detailed account of findings from both Word2Vec and GloVe embeddings. However, the discussion around these findings seems somewhat brief considering the complexity of the data. Could you expand on the potential societal implications of these biases in AI models? For instance, how might these biases influence real-world HR practices if these AI models are employed without correction?

Reviewers' comments:

Reviewer's Responses to Questions

**Comments to the Author**

1. If the authors have adequately addressed your comments raised in a previous round of review and you feel that this manuscript is now acceptable for publication, you may indicate that here to bypass the “Comments to the Author” section, enter your conflict of interest statement in the “Confidential to Editor” section, and submit your "Accept" recommendation.

Reviewer #3: (No Response)

Reviewer #4: (No Response)

2. Is the manuscript technically sound, and do the data support the conclusions?

Reviewer #3: Partly

Reviewer #4: Yes

3. Has the statistical analysis been performed appropriately and rigorously? 

Reviewer #3: I Don't Know

Reviewer #4: Yes

4. Have the authors made all data underlying the findings in their manuscript fully available?

Reviewer #3: No

Reviewer #4: Yes

5. Is the manuscript presented in an intelligible fashion and written in standard English?

Reviewer #3: Yes

Reviewer #4: Yes

6. Review Comments to the Author

**Reviewer #3**: This paper reports on a very interesting and potentially useful Natural Language Processing (NLP) analysis of mental health disorders in the context of employment. The methods appear to be appropriate although NLP is not my area of expertise. The results suggest that in language analysis, there is not a strong negative bias against mental health disorders. However, while this is interesting in theory, the authors do not provide sufficient argument as to how this would actually apply in the context of the hiring process. Unless engaged in supported employment, most job seekers with serious mental health conditions do not disclose to employers although employers might be able to discern mental health history from social media searches. Therefore, it is unclear what the results of NLP of mental health disorders and employability would look like in resume parsing. The utility of the results from an analysis of general text data rather than an analysis of resume and employment data analysis needs to be made more strongly especially as the findings may be biased by the general exclusion of people with serious mental health conditions from the workforce.

Specific comments

The summary of the literature on employment of people with serious mental illnesses shows some misunderstandings. Stigma is a barrier to employers hiring people with SMI, but not necessarily a barrier to wanting to work. Desire to work among adults with SMI in spite of individual, interpersonal, and structural barriers has been shown extensively. In addition, Supported Employment requires competitive employment as an outcome by definition; the two are not in opposition in the paper cited.

The paragraph describing how people with mental health conditions might be excluded by NLP needs more explanation and support: “NLPs are often trained to screen for a specific role, which often assumes a mentally health individual. Many people, due to psychiatric diagnosis or other attributes, revealed or not, do not fall into the desired category, however, could still perform the job well, if they were to be hired. These individuals will be missed by NLP models that are based on existing, ‘normal’ job descriptions. NLP models screening for a position might exclude candidates based on diagnosis, perhaps even with good reason, however, inappropriately for another.”

What is the evidence that people with disclosed or undisclosed psychiatric diagnoses would not fall into the desired category of mentally healthy? Evidence of social group identification from NLP has been given for race and gender categories, but not psychiatric disorders except for the studies cited which used self-disclosed diagnoses as an identifier. In the absence of identification of mental health disorders either through self-disclosure or language analysis, this logic remains only theoretical. This weakens the argument for the utility of the study.

To say that people with mental health disorders “could still perform the job well” and might be excluded “perhaps even with good reason” is opinion and biased language, as is the assumption that non-western or “non-conventionally written” resumes would be associated with mental illness.

Similarly, use of the word “normal” and “normality” as the contrast to mental illness is problematic and exclusionary. While a contrast needs to be made, the word “normal” for people without a disability is not inclusive language.

It is encouraging that in the analysis, there was not a strong negative association between mental health conditions and professions. However, it’s not clear that this absence of association in language would actually translate to less bias in resume parsing or employment. This lack of bias may reflect the profound exclusion of people with psychiatric disabilities from the labor force, rather than greater employability within the labor force. In other words, because people with psychiatric disabilities are generally excluded from employment, there would be less word embeddings of the two. This would also explain the finding that “more severe mental health disorders are more employable… than common mental health disorders” given that more common mental health disorders may occur in conjunction with employment more frequently in the data. Could this unexpected finding also be an artifact of the role of employed professionals in diagnosing and reporting on serious mental illness?

**Reviewer #4:** Since I've joined the reviewing in the second round, I'll review a bit differently than I would in the first round.

I think the authors do an excellent job in reviewing the related work and providing a solid backdrop. They also explain the methods that they have used well and do not shy away from taking a stance. In my opinion the reviewer comments from the first round have been addressed. However, one reviewer commented about the presentation of the results from the Word2Vec analogies, which has now been transformed into several tables. Nonetheless, it appears lazy, selective, and is hard to digest as a reader in this manner. I would therefore suggest that the authors perhaps combine analogies and top 5 or top 10 results into bigger tables with rows for the results and columns for the analogies (e.g. one table for professions, one table for employability attributes...).

Additionally, I struggle with the assertion made here "Word embeddings thus deliver a stable, reliable and valid estimate of biases." Aside from the fact that this strong conclusion follows a series of weak arguments with a lot of hedging ("often", "the notion", "can reflect"), it is later argued in the paper that word embeddings are (as most people in the field know) only as good at representing the world as the data they have been trained on and schizophrenia might show less bias because it is underrepresented in the data compared to depression (a good point to make). So how valid, reliable, and stable is this estimate?

The argument given for using word embeddings and GLoVE rather than more sophisticated quantitative methods ("an open-ended analogy approach allows for a wider scope of discovery") appears weak in light of not finding biases. In a first review round, I would have probably suggested extending the results with those from an alternative quantitative method to substantiate your claims.

Finally, two things struck me in the discussion on GLoVE embeddings. I'm by no means an expert on t-SNE and only have a vague notion on what the algorithm does, but I was wondering how sensible it is to compare the relative positions of any two points in this reduced space. I've always understood it more as a basic visualization technique that attempts to maintain clusters, so to what extent can two points be meaningfully compared? I may be off, but for me, the clusters that you colored green/blue/yellow/red are more intuitively represented in the space than the individual points.

Additionally, in this part of the discussion some word choices are made that are usually associated with statistical analyses while they seem to be made here on the basis of visual inspection (e.g., "marginally"). Perhaps consider phrasing this more clearly.

Spelling of author Nissim is incorrect in some places (Nisism) and at least once I've spotted psychosis as "psychois."

7. PLOS authors have the option to publish the peer review history of their article (what does this mean?). If published, this will include your full peer review and any attached files.

Reviewer #3: No

Reviewer #4: **Yes: **Franziska Burger

---

## [Author Response · Author response to Decision Letter 1]

5 Jun 2024

Rebuttal Letter

Journal Requirements

Editor: Please review your reference list to ensure that it is complete and correct. If you have cited papers that have been retracted, please include the rationale for doing so in the manuscript text, or remove these references and replace them with relevant current references. 

Answer: Is there a chance the editor could confirm the retracted reference they are referring to? After inspection of the list, we could not find a retracted article. We suspect that reference no. 17 in the last version of the manuscript (10.1093/schbul/sbn024) might have been a suspect, as there is a paper from the same lab, closely related and discussing similar findings (10.1080/09540260802564516) which indeed has been retracted, but which we never cited. If there is a different reference that has been retracted, please let us know and we will remove it. In any case, we removed reference no. 17 ( 10.1093/schbul/sbn024) due to comments by reviewer #3, more on that in the direct response to the reviewers.

Editor: In the methods section, you describe the use of Word2Vec and GloVe algorithms to assess bias in NLP models concerning employability. Can you clarify if these models were retrained with any specific modifications or if they were used as pre-trained models directly? 

Answer: The models were used as they were, directly downloaded from 'word2vec/heavy/GoogleNews-vectors-negative300', download_dir='.magnitude/' and 

http://magnitude.plasticity.ai/glove/light/glove.6B.300d.magnitude

with no re-training. We added a sentence regarding this information in the sections ‘2.3 Word2Vec’ and ‘2.4 Global Vectors for Word Representation (GloVe)’

Editor: Furthermore, how did you ensure that the analogy tasks and cosine similarity measures were sufficiently sensitive and specific to detect subtle forms of bias that might not be overt?

Answer: Word analogies and cosine similarity have indeed limitations in detecting complex biases, and there are more sophisticated methods to detect subtle forms of bias. This has been discussed in our section ‘2.5.1 Alternative Methods’. However, these methods were chosen for their established use in bias research and their interpretability in our specific context, as well as building on work by Straw and Callison-Burch (2020). As mentioned in our section ‘4.2 Future research’ while our results show limited overt bias, we recognize the need for more nuanced techniques like WEAT or SEAT to uncover subtle biases. Future work will explore these methods to build upon our findings. More on this in our response to reviewer #4.

Editor: The results section provides a detailed account of findings from both Word2Vec and GloVe embeddings. However, the discussion around these findings seems somewhat brief considering the complexity of the data. Could you expand on the potential societal implications of these biases in AI models? For instance, how might these biases influence real-world HR practices if these AI models are employed without correction?

Answer: We have covered, at length, the topic of how these biases influence real-world HR practices if these AI models are employed without correction in our sections “1.3.1 Recruiting and Occupational Bias in NLP” and “1.3.2 Job Recommender Systems and Occupational Bias in NLP”. We do not see the need to elaborate on this in the discussion.

Reviewers

Reviewer #3: The results suggest that in language analysis, there is not a strong negative bias against mental health disorders. However, while this is interesting in theory, the authors do not provide sufficient argument as to how this would actually apply in the context of the hiring process. Unless engaged in supported employment, most job seekers with serious mental health conditions do not disclose to employers although employers might be able to discern mental health history from social media searches. Therefore, it is unclear what the results of NLP of mental health disorders and employability would look like in resume parsing. The utility of the results from an analysis of general text data rather than an analysis of resume and employment data analysis needs to be made more strongly especially as the findings may be biased by the general exclusion of people with serious mental health conditions from the workforce.

Answer: We agree that job seekers with serious mental health conditions often do not disclose to employers in their resumes. What we are saying is: Our findings should make job seekers with mental health disorders hopeful, as they could openly disclose their condition to employers without facing discrimination. We have added this information to our conclusion.

Of course this might be the result of not enough people disclosing it, which in turn results in fewer mentioning in the initial corpus and thus limited discrimination, i.e. there is a chance this lack of discrimination is not a result of changing attitude towards mental health disorders on the side of employers, but rather, as discussed in this and the last revision round as well as in our discussion section, it just might be the lack of mentioning in the training corpus. This would still not change the fact that there appears to be low bias.

Reviewer #3: The summary of the literature on employment of people with serious mental illnesses shows some misunderstandings. Stigma is a barrier to employers hiring people with SMI, but not necessarily a barrier to wanting to work. Desire to work among adults with SMI in spite of individual, interpersonal, and structural barriers has been shown extensively. In addition, Supported Employment requires competitive employment as an outcome by definition; the two are not in opposition in the paper cited.

Answer: That is true. We amended the corresponding paragraph. 

Reviewer #3: What is the evidence that people with disclosed or undisclosed psychiatric diagnoses would not fall into the desired category of mentally healthy? Evidence of social group identification from NLP has been given for race and gender categories, but not psychiatric disorders except for the studies cited which used self-disclosed diagnoses as an identifier. In the absence of identification of mental health disorders either through self-disclosure or language analysis, this logic remains only theoretical. This weakens the argument for the utility of the study.

Answer: This is a very similar argument to a query raised by reviewer #1 in the previous round. As discussed last time, and as reviewer #3 points out correctly, there are no investigations into writing style of people with mental health disorders (this would be an excellent idea for a future paper). 

What we argue, instead, is the following: Specific word use patterns of people with mental illness have been identified, while detection of general mental illnesses and even reliable estimation of population mental health is possible through analyses of social media posts, surveys, narrative writings, and interviews. At the same time, deep learning models can detect social media users at risk for developing a mental disorder, deduced from online posts represented with linguistic features at different levels, including messages and corresponding writing style and emotions conveyed. 

Contrary to what reviewer #3 states, the studies cited (predominantly those from our section ‘1.3.1 Recruiting and Occupational Bias in NLP’ ) do not use self-disclosed diagnoses as an identifier. 

Hence, even though no study has looked at resume writing by people with mental health disorders, we argue, the chances of an NLP algorithm being able to detect differences in writing styles between healthy people and people with mental health disorders would be extremely high. 

Reviewer #3: To say that people with mental health disorders “could still perform the job well” and might be excluded “perhaps even with good reason” is opinion and biased language, as is the assumption that non-western or “non-conventionally written” resumes would be associated with mental illness. Similarly, use of the word “normal” and “normality” as the contrast to mental illness is problematic and exclusionary. While a contrast needs to be made, the word “normal” for people without a disability is not inclusive language. While a contrast needs to be made, the word “normal” for people without a disability is not inclusive language.

Answer: We apologise for this oversight; we have now followed American Psychological Association guidelines to refer to people experiencing mental illness with bias free language.

We have amended the paragraph, changing ‘normal’ to ‘healthy’. 

We did not state that we assume non-western or non-conventionally written resumes would be associated with mental illness. We state that NLP algorithms trained on data of healthy individuals, when encountering a non-western or non-conventionally written resume or cover letter, might mistakenly assume these resumes as written by people experiencing mental illness, only because it is different to the training data. This is speaking to the un-inclusive manner of training data going into algorithms, not to an association between non-western writing style and mental illness. Hence, we called it a false-positive scenario for exclusion, i.e. the algorithm mistaking anything that is not what it was trained on gets automatically excluded.

Reviewer #3: It is encouraging that in the analysis, there was not a strong negative association between mental health conditions and professions. However, it’s not clear that this absence of association in language would actually translate to less bias in resume parsing or employment. This lack of bias may reflect the profound exclusion of people with psychiatric disabilities from the labor force, rather than greater employability within the labor force. In other words, because people with psychiatric disabilities are generally excluded from employment, there would be less word embeddings of the two. This would also explain the finding that “more severe mental health disorders are more employable… than common mental health disorders” given that more common mental health disorders may occur in conjunction with employment more frequently in the data. Could this unexpected finding also be an artifact of the role of employed professionals in diagnosing and reporting on serious mental illness?

Answer: This is a very good point, which, however, had been addressed in our discussion.

This paper was not an investigation into employability of people with mental health disorders, hence our findings do not comment on employability of this group in the labour force. Our findings only reflect how the word2vec algorithm estimates the employability of people with mental health disorders. As, at the moment, they seem to not be discriminated by the word2vec algorithm, they might be encouraged to openly disclose their condition. We addressed the point that, because people with psychiatric disabilities are generally excluded from employment, there would be less word embeddings of the two, in our discussion section. This, however, does not change the fact that the algorithm appears to hold low bias against these individuals. The algorithm might thus do the right thing for the wrong reason. 

What is true is a potential scenario as follows: 

Supposedly more people disclose their mental health disorder, still get hired. Algorithms would be newly trained on data featuring individuals with mental health disorders. There are two extreme scenarios: Either they perform just as good, or even better than their health peers, which would then in turn lead to even less discrimination in the algorithm. 

The second option would be people with mental health disorders performing worse than healthy people, thus getting excluded from the workforce, thus resulting periodically in an algorithm that has learned to exclude mentally ill people. However, after some time, through their consequent absence in the training data, when retraining the algorithm again at a later point in time, they are again not featured in the training corpus, i.e. the algorithm again has low bias against these individuals and the cycle starts again. This seems to be the point reviewer #3 tries to make. 

We do not know what will happen, only that in both cases, the algorithm seems to not discriminate against people with mental health disorders, if potentially for the wrong reasons. 

Reviewer #4: One reviewer commented about the presentation of the results from the Word2Vec analogies, which has now been transformed into several tables. Nonetheless, it appears lazy, selective, and is hard to digest as a reader in this manner. I would therefore suggest that the authors perhaps combine analogies and top 5 or top 10 results into bigger tables with rows for the results and columns for the analogies (e.g. one table for professions, one table for employability attributes...).

Answer: As reviewer #4 states here, another reviewer in the previous revision round already gave us feedback on our method of results presentation, with which we complied. This new suggestion now seems to be a personal style preference and we believe our method of presentation as it currently stands, as done in compliance with the earlier reviewer, is sufficient in presenting our results in a clear and understandable way for a reader.

Reviewer #4: Additionally, I struggle with the assertion made here "Word embeddings thus deliver a stable, reliable and valid estimate of biases." Aside from the fact that this strong conclusion follows a series of weak arguments with a lot of hedging ("often", "the notion", "can reflect"), it is later argued in the paper that word embeddings are (as most people in the field know) only as good at representing the world as the data they have been trained on and schizophrenia might show less bias because it is underrepresented in the data compared to depression (a good point to make). So how valid, reliable, and stable is this estimate?

Answers: 

Regarding the notion that a strong conclusion follows a series of weak arguments: We do not follow the reasoning that the arguments are weak. We refuted and disproved arguments and methodological flaws by other researchers, who misinterpreted earlier research. 

Regarding the notion that there is a lot of hedging: We do not see a problem with that.

Regarding the notion that it is later argued in the paper that word embeddings are (as most people in the field know) only as good at representing the world as the data they have been trained on and schizophrenia might show less bias because it is underrepresented in the data compared to depression: As we keep saying in our rebuttal letters, we do not question that there are likely few mentions of schizophrenia in the training data and thus probably inaccurate word embeddings as a result. Still, the model does not discriminate against schizophrenia. 

Regarding the question ‘So how valid, reliable, and stable is this estimate?’: See query above

Reviewer #4: The argument given for using word embeddings and GLoVE rather than more sophisticated quantitative methods ("an open-ended analogy approach allows for a wider scope of discovery") appears weak in light of not finding biases.

Answer: The absence of detected biases in our results should not be interpreted as a weakness of the method, but rather indicative of the possible characteristics of the underlying data, which may suggest a lack of overt discriminatory bias within the corpus used. Denouncing our method as weak because no bias was found is similar to calling a blood test inaccurate only because it does not find disease within a healthy patient. Future enhancements, as outlined, will include more nuanced methods such as WEAT, further refining our ability to detect and interpret subtle biases in employment-related contexts concerning mental health.

Reviewer #4: Finally, two things struck me in the discussion on GLoVE embeddings. I'm by no means an expert on t-SNE and only have a vague notion on what the algorithm does, but I was wondering how sensible it is to compare the relative positions of any two points in this reduced space. I've always understood it more as a basic visualization technique that attempts to maintain cl

---

## [Decision Letter · Decision Letter 2]

21 Aug 2024

PONE-D-24-00286R2­­­­Schizophrenia more employable than depression? Language-based artificial intelligence model ratings for employability of psychiatric diagnoses and somatic and healthy controlsPLOS ONE

Dear Dr. Lange,

Thank you for submitting your manuscript to PLOS ONE. After careful consideration, we feel that it has merit but does not fully meet PLOS ONE’s publication criteria as it currently stands. Therefore, we invite you to submit a revised version of the manuscript that addresses the points raised during the review process.

We look forward to receiving your revised manuscript.

Kind regards,

Sumeet Kaur Sehra

Academic Editor

PLOS ONE

Journal Requirements:

Reviewers' comments:

Reviewer's Responses to Questions

**Comments to the Author**

1. If the authors have adequately addressed your comments raised in a previous round of review and you feel that this manuscript is now acceptable for publication, you may indicate that here to bypass the “Comments to the Author” section, enter your conflict of interest statement in the “Confidential to Editor” section, and submit your "Accept" recommendation.

Reviewer #3: (No Response)

Reviewer #4: (No Response)

2. Is the manuscript technically sound, and do the data support the conclusions?

Reviewer #3: Yes

Reviewer #4: Yes

3. Has the statistical analysis been performed appropriately and rigorously? 

Reviewer #3: Yes

Reviewer #4: Yes

4. Have the authors made all data underlying the findings in their manuscript fully available?

Reviewer #3: No

Reviewer #4: Yes

5. Is the manuscript presented in an intelligible fashion and written in standard English?

Reviewer #3: Yes

Reviewer #4: Yes

6. Review Comments to the Author

Reviewer #3: (No Response)

Reviewer #4: While the authors have addressed most comments to my complete satisfaction, the presentation of the results can still be improved. I agree that the authors addressed this in response to Reviewer #1. However, R1 did not do a second round of review, and it is unknown whether R1 is happy with the new presentation. I still find it difficult to draw conclusions from it and while my suggestion may be personal and does absolutely not have to be followed in that format (it was just a suggestion to save the authors time), I do invite the authors to reconsider their presentation because I stand by my original comment that it appears lazy (no further explanation or directing of attention in text, only printed tables that are never referenced in the text), selective (why these tables and not others that were moved to supplementary materials?), and is not very useful to the reader in this manner (what are you trying to get the reader to see with this selection? and are you achieving it?).

Other comments were more minor and have been sufficiently addressed.

Minor comment in response to rebuttal: I never said the method is weak in light of not finding results, I said the argument for using only this approach is weakened in light of not finding results and adding a second approach capable of uncovering more subtle biases would increase the utility of the paper. I do agree that this has been sufficiently discussed in the limitations and future work sections, though.

7. PLOS authors have the option to publish the peer review history of their article (what does this mean?). If published, this will include your full peer review and any attached files.

Reviewer #3: No

Reviewer #4: No

---

## [Author Response · Author response to Decision Letter 2]

28 Aug 2024

Rebuttal Letter

Journal Requirements

Editor: Please review your reference list to ensure that it is complete and correct. If you have cited papers that have been retracted, please include the rationale for doing so in the manuscript text, or remove these references and replace them with relevant current references. 

Answer: As stated in our last rebuttal letter that featured this exact comment, we kindly ask the editor to confirm the retracted and/or missing reference they are referring to, since, after inspection of our list, we could not find a retracted article and we believe our reference list is complete and correct.

Reviewers

Reviewer #4: I still find it difficult to draw conclusions from [the presentation of the result] and while my suggestion may be personal and does absolutely not have to be followed in that format (it was just a suggestion to save the authors time), I do invite the authors to reconsider their presentation because I stand by my original comment that it appears lazy (no further explanation or directing of attention in text, only printed tables that are never referenced in the text), selective (why these tables and not others that were moved to supplementary materials?), and is not very useful to the reader in this manner (what are you trying to get the reader to see with this selection? and are you achieving it?).

Answer: 

The reviewer makes four different arguments regarding presentation of results, specifically the tables. 

First, “no further explanation or directing of attention in text, only printed tables that are never referenced in the text”. 

We have now added text in the result section referencing to all tables used in that section.

Second, the reviewer thinks our presentation of the results appears “selective (why these tables and not others that were moved to supplementary materials?”. 

In earlier versions that were submitted to the journal we included all tables. However, reviewer #1 advised us to move some to the supplementary material. We do accept the critique that it appears selective. There is no order to the inclusion of the findings, or to selection of which went to the supplementary material. This is since they all investigate very similar queries (mental health and employability). Further, none but one analogy table featured bias in the top-1 returned word. We included that table in the main manuscript. We added text regarding this in the newly added text in the results section.

Third, the reviewer critiques our presentation of the results to be “not very useful to the reader in this manner (what are you trying to get the reader to see with this selection, and are you achieving it?” 

We added new text to the methods section 2.3 Word2Vec, stating that we select job titles and fields that are associated with prestige and status, e.g., medicine, law, engineering, finance. Competition to enter such occupations is hence higher than most jobs. This, we argue, would result in easily detectable potential bias against mental illness, is often aimed at ability. We are showing the reader that in the word embeddings examined, analogies did not contain biased words within the top-10 returned items, indicating that the model might see people with mental illness as equally able to enter and/or perform in high earning professions. We have added a statement explaining this in the results section.

The reviewer is further alluding to a comment from the last round of reviews, suggesting to “combine analogies and top 5 or top 10 results into bigger tables with rows for the results and columns for the analogies”.

Doing this, we believe, would complicate reading and understanding of the reader as then they would be bombarded with large tables of over 10 analogies. 

We hope we have found a middle ground with the suggestions of the reviewer and our own views regarding presentation of the results. We appreciate the continuous feedback and commitment to the improvement of this manuscript by all reviewers.

---

## [Decision Letter · Decision Letter 3]

2 Dec 2024

­­­­Schizophrenia more employable than depression?

Language-based artificial intelligence model ratings for employability of psychiatric diagnoses and somatic and healthy controls

PONE-D-24-00286R3

Dear Dr. Lange,

We’re pleased to inform you that your manuscript has been judged scientifically suitable for publication and will be formally accepted for publication once it meets all outstanding technical requirements.

Kind regards,

Sumeet Kaur Sehra

Academic Editor

PLOS ONE

Additional Editor Comments (optional):

Reviewers' comments:

Reviewer's Responses to Questions

**Comments to the Author**

1. If the authors have adequately addressed your comments raised in a previous round of review and you feel that this manuscript is now acceptable for publication, you may indicate that here to bypass the “Comments to the Author” section, enter your conflict of interest statement in the “Confidential to Editor” section, and submit your "Accept" recommendation.

Reviewer #4: All comments have been addressed

2. Is the manuscript technically sound, and do the data support the conclusions?

Reviewer #4: Yes

3. Has the statistical analysis been performed appropriately and rigorously? 

Reviewer #4: Yes

4. Have the authors made all data underlying the findings in their manuscript fully available?

Reviewer #4: Yes

5. Is the manuscript presented in an intelligible fashion and written in standard English?

Reviewer #4: Yes

6. Review Comments to the Author

Reviewer #4: Yes, we've reached a middle ground. Thank you for a pleasant reviewing process, congrats to a good manuscript, and I wish you all the best in your future research endeavors.

7. PLOS authors have the option to publish the peer review history of their article (what does this mean?). If published, this will include your full peer review and any attached files.

Reviewer #4: No

---

## [Editor Report · Acceptance letter]

6 Dec 2024

PONE-D-24-00286R3 

PLOS ONE

Dear Dr. Lange, 

I'm pleased to inform you that your manuscript has been deemed suitable for publication in PLOS ONE. Congratulations! Your manuscript is now being handed over to our production team.

Kind regards, 

on behalf of

Dr. Sumeet Kaur Sehra 

Academic Editor

PLOS ONE